# Analyses of the Global Multilocus Genotypes of the Human Pathogenic Yeast *Cryptococcus neoformans* Species Complex

**DOI:** 10.3390/genes13112045

**Published:** 2022-11-06

**Authors:** Megan Hitchcock, Jianping Xu

**Affiliations:** Department of Biology, McMaster University, Hamilton, ON L8S 4K1, Canada

**Keywords:** *Cryptococcus*, *Cryptococcus neoformans*, multilocus sequence typing, sequence type, yeast, geographical distribution, recombination

## Abstract

*Cryptococcus neoformans* species complex (CNSC) is a globally distributed human opportunistic yeast pathogen consisting of five major molecular types (VNI, VNII, VNB, VNIII and VNIV) belonging to two species, *C. neoformans* (VNI, VNII and VNB, collectively called serotype A) and *C. deneoformans* (VNIV, commonly called serotype D), and their hybrids (VNIII, serotype AD). Over the years, many studies have analyzed the geographical distribution and genetic diversity of CNSC. However, the global population structure and mode of reproduction remain incompletely described. In this study, we analyze the published multilocus sequence data at seven loci for CNSC. The combined sequences at the seven loci identified a total of 657 multilocus sequence types (STs), including 296 STs with known geographic information, representing 4200 non-redundant isolates from 31 countries and four continents. Among the 296 STs, 78 and 52 were shared among countries and continents, respectively, representing 3643 of the 4200 isolates. Except for the clone-corrected serotype D sample among countries, our analysis of the molecular variance of the 4200 isolates revealed significant genetic differentiations among countries and continents in populations of CNSC, serotype A, and serotype D. Phylogenetic analyses of the concatenated sequences of all 657 STs revealed several large clusters corresponding to the major molecular types. However, several rare but distinct STs were also found, representing potentially novel molecular types and/or hybrids of existing molecular types. Phylogenetic incompatibility analyses revealed evidence for recombination within all four major molecular types—VNI, VNII, VNIV and VNB—as well as within two VNB subclades, VNBI and VNBII, and two ST clusters around the most common STs, ST5 and ST93. However, linkage disequilibrium analyses rejected the hypothesis of random recombination across most samples. Together, our results suggest evidence for historical differentiation, frequent recent gene flow, clonal expansion and recombination within and between lineages of the global CNSC population.

## 1. Introduction

The human pathogenic *Cryptococcus* (HPC) is a group of globally distributed basidiomycete yeasts. These yeasts are opportunistic pathogens to humans and other mammals. They are commonly found in soil, avian excretion and rotting tree barks [1,2,3,4]. HPC consists of two species complexes, the *Cryptococcus neoformans* species complex (CNSC) and the *Cryptococcus gatti* species complex (CGSC). Globally, most human infections are caused by strains of CNSC. CNSC tends to infect immunocompromised hosts and is a leading cause of death in HIV patients [5,6]. Infections can lead to systemic cryptococcosis, with the most common and detrimental form being cryptococcal meningitis. In 2014, there were ~223,100 recorded cases of cryptococcal meningitis resulting in ~180,100 deaths worldwide [5,6]. With cases on the rise over the past five decades due to increasing populations of immunocompromised hosts [6,7], it is important that we improve our understanding of the global distribution and genetic diversity of *C. neoformans.*

CNSC is a highly heterogeneous group of organisms, with divergent lineages showing >10% nucleotide sequence divergence [1]. Over the last 50 years, a variety of molecular markers have been used to identify strains of CNSC [8]. These markers have revealed divergent lineages within CNSC. The current emerging consensus separates CNSC into two species, *C. neoformans* (serotype A) and *C. deneoformans* (serotype D). *C. neoformans* (Serotype A) is further divided into three major molecular types, VNI, VNB and VNII, while *C. deneoformans* (serotype D) corresponds to the molecular type VNIV. In addition to these four major molecular types, VNB was further divided into two subtypes, VNBI and VNBII, and diploid/aneuploid hybrids have been observed in nature and are referred to as VNIII or serotype AD hybrids [1,8,9,10].

To help standardize the genotyping system and make it easy to share information among labs, in 2007, the International Society for Human and Animal Mycology (ISHAM) established a committee to set up a multi-locus sequence typing (MLST) method. The recommended system was published in 2009 and included partial DNA sequences at the following seven loci: *CAP59, GPD1, LAC1, PLB1, SOD1, URA5* and *IGS1* [11]. Subsequently, an online international fungal multi-locus serotyping database (IFMLST) was established for storing and comparing the MLST data. This data repository is comprised of allelic profiles for each recorded sequence type (ST) and the nucleotide sequence for each determined allele type (AT) at each of the seven loci [12]. Newly genotyped strains can be compared to the database to determine their AT and ST profiles. The MLST system is a great tool for the consistent and efficient comparison of strain genotypes across labs. However, little analysis has been conducted utilizing the publicly available dataset.

Considering the high global burden of infections by CNSC, it is important to understand the global population genetic variation of this species complex. In this paper, we investigate the global genotype distribution and population structure by analyzing the 4200 CNSC isolates with MLST data published in 41 studies. We aim to estimate the overall geographical pattern of genetic variation and determine if recombination plays a role in shaping the diversity observed within individual species and individual molecular types of CNSC. We hypothesize that the genetic variations within CNSC are geographically structured, and recombination plays an important role during the evolution of this species complex.

## 2. Materials and Methods

### 2.1. Strains, Genotypes and Metadata

We conducted a literature search through PubMed using the search term “Cryptococcus MLST”. All retrieved papers were scanned for isolates, with MLST data belonging to *C. neoformans*, *C. neoformans* var. *grubii*, *C. neoformans* var*. neoformans*, *C. neoformans* serotype A and *C. neoformans* serotype D. The reported MLST data on the isolates of CNSC from these studies were extracted and compiled. For each sequence type, its allelic profile and DNA sequence data at the seven loci were retrieved from the publicly available International Fungal Multi Locus Sequence Typing Database. All MLST data for strains of CNSC published by January 2022 were included. A total of 4200 isolates from 41 studies were included in our population genetic analyses. The detailed descriptions of these isolates and their genotype data, including their protocols for DNA extraction, amplification and sequencing and allelic profiling can be found in the original reports [13,14,15,16,17,18,19,20,21,22,23,24,25,26,27,28,29,30,31,32,33,34,35,36,37,38,39,40,41,42,43,44,45,46,47,48,49,50,51,52,53].

### 2.2. Phylogenetic Distribution of Strains and Genotypes

To determine the relationships among STs within CNSC, DNA sequences at the seven MLST loci were concatenated based on the allelic profile for each sequence type. The concatenated sequences were then imported into MEGA X and aligned through Muscle [54,55]. A phylogenetic tree was reconstructed using a neighbor-joining method based on the K2P distance model (Kimura 2-parameter model) [54]. Associated molecular type classifications from the IFMLST database, as well as geographical and ecological source data based on the published literature, were added to each ST to show their distributions on the phylogenetic tree using the iTOL software [56].

### 2.3. Population Genetic Analyses

To investigate if geographic populations of CNSC were genetically subdivided, strains were separated into different countries and continents based on their metadata and analyzed using the GenAlEx V.6.5 program [57]. In our population genetic analyses, three taxonomy group-based samples were analyzed: the entire CNSC sample, the *C. neoformans* (serotype A) sample and the *C. deneoformans* (serotype D) sample. For each of the three taxonomic samples, two types of datasets were analyzed: non-clone-corrected (NCC), and clone-corrected (CC). The NCC datasets included all isolates extracted from the literature. For the CC dataset, only one representative strain of each ST from each country was included in our analyses.

For each of the six datasets (three taxonomic NCC datasets and three taxonomic CC datasets), we separately conducted the analyses of molecular variance (AMOVA) at the country and continental levels. In addition, Wright’s F_ST_ values were obtained between pairs of geographic samples. To reduce potential biases due to small samples sizes, individual subpopulations with <five isolates were excluded from the country analysis. Statistical significance for each test was obtained by comparing the observed with the distributions of 1000 permutated datasets generated based on a null hypothesis of no genetic differentiations within each analyzed dataset.

The MLST dataset including all 657 STs was also used to identify potential evidence for recombination within individual species (serotypes), molecular types and selected phylogenetic clusters. For this test, only the clone-corrected allelic profiles at the seven loci were analyzed. Specifically, phylogenetic compatibility and linkage disequilibrium analyses were performed with 1000 randomizations for the total clone-corrected CNSC dataset, the serotype A subset, the serotype D subset, the three molecular types (VNI, VNII and VNB) of serotype A, two subtypes of VNB (VNBI and VNBII) as well as phylogenetic clusters associated with the two most common STs to investigate potential evidence for recombination across CNSC. Details regarding the underlying principles of these tests and how these tests were conducted can be found in the MultiLocus V1.3 manual [58].

## 3. Results

As of January 2022, there were 657 total sequence types (STs) deposited into the *Cryptococcus* MLST database for CNSC, with associated DNA sequence data for all seven loci. Of the 657 STs, the geographical location information was documented for 296 (45%), while the remaining 361 (55%) STs had unknown geographic information. Our population genetic analyses focused on the 296 STs. The 296 STs represented 4200 CNSC isolates, as extracted from 41 published reports. The metadata for all isolates were retrieved from these published reports. Below, we summarize the retrieved data on the 4200 isolates and present the results of our analyses.

### 3.1. Geographical and Ecological Distributions

The geographic distribution of the non-redundant 4200 CNSC isolates is presented in Table 1. These isolates were from 31 countries and five continents, with the majority being found in Asia (61.9%), followed by Africa (12.6%), Europe (14.3%), South America (10.9%) and North America (0.3%). At the country level, the highest number of isolates in this dataset came from China (1216 isolates; 28.95%), while the lowest came from the Congo, the Dominican Republic and the Democratic Republic of Congo (with one isolate each). In between these two extremes, the second largest national population of CNSC in the retrieved dataset was from Thailand (524 isolates; 12.27%), followed by India (380; 9.05%), Brazil (318 isolates; 7.57%), South Africa (268 isolates; 6.38%), Uganda (241 isolates; 5.74%), France (226; 5.38%), Italy (151 isolates; 3.60%), Germany (145 isolates, 3.45%), Vietnam (136 isolates; 3.24%) and Japan (119 isolates; 2.83%). The remaining 20 countries each had <100 isolates analyzed, and, together, they contributed 476 isolates to the analyzed dataset. The geographic associations of the 296 STs are presented in Appendix A. Among the 296 STs, ST5 was the most abundant; it was found across 18 countries on four continents (Appendix A).

Of the 657 STs in the MLST database, only 284 had ecological niche information (Appendix A; Table 2). These 284 STs represented a total of 4064 isolates, while the remaining 373 STs had no ecological niche/source data. Here, we broadly categorized the isolates into three ecological sources: clinical, environmental and veterinary. The majority of the 4200 isolates were collected from clinical sources (3370 isolates; 80.24%), followed by environmental (648 isolates; 15.43%), and veterinary (46 isolates; 1.10%) sources, leaving 3.24% of isolates with unknown source information. The ecological distributions of the individual STs are shown in Appendix A. A total of 14 STs were found in all three niches; 3 STs were found from both clinical and veterinary sources only; 32 STs were found in both clinical and environmental sources only; and no ST was shared between only environmental and veterinary sources. The remaining 235 STs with ecological niche information were each found in only one of the three ecological niches (Table 2).

Table 2 summarizes the geographic and ecological distributions of the 296 STs in the published MLST literature for CNSC. Geographically, among the 296 STs, 15 STs (representing a total of 2675 isolates) were found in all four continents, 9 STs (representing a total of 656 isolates) were found in three of the four continents, 28 STs (representing a total of 312 isolates) were found in two of the four continents and 244 (representing 557 isolates) were found in only one of the continents (Table 2). Among the 244 STs, 176 were each represented by only one isolate in the database. Ecologically, among the 296 STs, 284 STs, including 4064 isolates, had ecological niche data. Of these 284 STs, 14 (representing 2550 isolates) were found in all three ecological niches, 35 STs (representing 1034 isolates) were found in two of the three ecological niches and 235 STs (representing 480 isolates) were found in one niche only (Table 2). The detailed geographic and ecological distributions for each of the 296 STs are shown in Appendix A. At the country level, 78 (26%) sequence types were reported from two or more countries each (Appendix A).

### 3.2. DNA Sequence Variation

The allelic profiles of each ST, including the allele type (AT) number at each of the seven MLST loci (*CAP59, GPD1, LAC1, IGS1, PLB1, SOD1, URA5*), were retrieved for all 657 STs in the database. Summaries of the allele types across the total sample, the serotype A sample and the serotype D sample are shown in Table 3. The differences in length of bp per allele type for each gene range from 0 difference for *CAP59* to 45bp difference for *IGS1.* Among the seven loci, in the total sample, *PLB1* had the fewest number of alleles (44) while *IGS1* had the most (93). A largely similar pattern was observed for the serotype A and serotype D samples, where *IGS1* had the highest allele number in both. However, *GPD1* had the lowest allele number in the serotype D sample. The range of occurrence of each allele type in each of the samples is shown in Appendix A.

### 3.3. Phylogenetic Analysis

The *C. neoformans* species complex is commonly grouped into five broad molecular types: VNI, VNII, VNIII, VNB and VNIV. The strains of VNI, VNII and VNB belonged to *C. neoformans* (serotype A); the strains of VNIV belonged to *C. deneoformans* (serotype D); and the strains of VNIII (serotype AD) represented hybrids of serotypes A and D. For some VNB strains, they were further classified into VNBI and VNBII [8,10]. The molecular type designations were mostly based on the restriction enzyme digest pattern of the *URA5* sequence, amplified fragment length polymorphisms or PCR fingerprinting [8]. Analyses of the concatenated sequences at the seven MLST loci showed a largely consistent clustering of STs into their original molecular type designations, with VNIV being the most distant from VNI, VNII and VNB (Figure 1). Similarly, except for one ST (ST434) that was originally assigned to VNBII but was clustered more closely with VNBI strains, all other STs originally assigned to VNBI and VNBII were clearly separated into two groups (Figure 1). However, there were several other notable inconsistencies. Specifically, six STs originally assigned to VNIV (ST521, ST254, ST266, ST355, ST489 and ST538) showed a closer relationship with the VNI clade. In contrast, 14 STs originally assigned to VNI (ST210, ST224, ST225, ST249, ST259, ST263, ST326, ST345, ST353, ST354, ST358, ST365, ST366 and ST651) and three STs originally assigned to VNII (ST221, ST222 and ST363) showed intermediate phylogenetic placing between the major serotypes A and D genotypes. Interestingly, 15 of the above 23 STs contained alleles with mixed clustering patterns, where some of the alleles belonged to the serotype A cluster, while others belonged to the serotype D allele cluster (Table 4; Appendix A). In addition, multiple STs originally assigned to molecular types VNI, VNII and VNB showed ambiguous placements within serotype A, often showing large distances from the three main molecular types (Figure 1). In contrast, 51 STs with previously undefined molecular type assignments were grouped into various species/molecular types (Figure 1). Phylogenic trees showing relationships among allele sequences for each of the seven genes can be seen in Appendix A.

### 3.4. AMOVA

Because of the highly skewed population sizes among countries, with seven countries each having fewer than five isolates represented, our AMOVA was conducted separately at the continental and country levels, instead of through a two-level hierarchical analysis. At the country level, only those with more than five isolates represented are included. The overall objective of our AMOVA was to assess how much geographic separations contributed to the total genetic variation. Below, we briefly summarize the results.

At the continental level analyses, in the none-clone-corrected sample, genetic variations within continents contributed 72%, 78% and 84% of the total observed genetic variations in the total CNSC population, the serotype A population and the serotype D population, respectively. The remaining 28%, 22% and 16% were attributed among continents. The within-continent and among-continent contributions for each of the three taxonomic populations were statistically significant at the *p* < 0.001 level (Table 5). In the three clone-corrected samples, genetic variations within continents contributed 96%, 98% and 96% of the total observed genetic variations in the entire CNSC population, the serotype A population and the serotype D population, respectively. These percentages were significantly greater than those without clone corrections. The remaining 4%, 2% and 4% were attributed among continents. Despite the smaller percentages of contributions, the among-continent contributions for two of the three population types were statistically significant at *p* < 0.001, while the serotype D population was significant at *p* = 0.03 (Table 5). The pairwise comparisons between continents for the three taxonomic samples are shown in Table 6.

At the country level, in the non-clone-corrected sample analyses, genetic variations within countries contributed 39%, 41% and 74% of the total observed genetic variations in the total CNSC sample, the serotype A sample and the serotype D sample, respectively. The remaining 61%, 59% and 26% were attributed among countries. The within-country and among-country contributions for each of the three taxonomic sample types were statistically significant at the *p* < 0.001 level (Table 7). In the three clone-corrected samples, genetic variations within countries contributed 83%, 79% and 99% of the total observed genetic variations in the total CNSC sample, the serotype A sample and the serotype D sample, respectively. Similar to those observed at the continental level, these percentages by within-countries in the clone-corrected samples were significantly greater than those without clone corrections. The remaining 17%, 21% and 1% were attributed among countries. Despite the smaller percentages of contributions, except for the serotype D sample, the remaining two among-country contributions for the three sample types were statistically significant at the *p* < 0.001 level (Table 7). The pairwise comparisons among countries for the three taxonomic samples are shown in Appendix A.

### 3.5. Recombination & Linkage Disequilibrium

We investigated the potential signatures of recombination among different samples of CNSC using two common indicators: phylogenetic incompatibility and linkage equilibrium. Here, aside from the three large taxonomic samples (the total CNSC, serotype A and serotype D), we also separately analyzed the three major molecular types (VNI, VNII and VNB) within serotype A, two subclades (VNBI and VNBII) within VNB as well as the two genotype clusters closely related to the two most dominant STs (ST5 and ST93) in the global sample.

In the phylogenetic incompatibility test, we found that none of the 10 samples showed 100% phylogenetic compatibility (Table 8). Specifically, 4 (the total CNSC, serotype A, serotype D and clade VNI) of the 10 analyzed datasets showed no phylogenetic compatibility among the seven loci, which was consistent with the evidence of recombination among all 21 pairs of loci within each of the four samples. For VNII-, VNBI- and ST5-associated genotype groups, 16 of the 21 pairwise loci combinations were phylogenetically incompatible, with only five pairs (23.8%) being phylogenetically compatible. For the VNB and VNBII datasets, 19 of the 21 pairwise loci were phylogenetically incompatible. For the ST93-associated genotype group, 5 of the 21 pairs showed phylogenetic incompatibility, which was also consistent with the evidence for recombination in this sample.

Linkage disequilibrium analyses revealed that in nine of the ten samples, the null hypothesis of random recombination was rejected (Table 8). The only exception was the ST93-associated genotype group, where the null hypothesis of random recombination was not rejected, likely due to the small sample size and the lack of statistical power to reject the null hypothesis. However, variable numbers of pairs of loci within each of the ten samples showed no significant deviation from those expected under the random recombination hypothesis (Appendix A). For example, in the VNII sample, 4 of the 21 loci pairs had observed genotype frequencies not significantly different from random recombination, with all 4 involving the *IGS1* locus. In the VNB sample, 9 of the 21 loci pairs had observed genotype frequencies not significantly different from random recombination. Interestingly, while no evidence for linkage equilibrium across all 21 loci pairs was observed in the ST93-associated genotype cluster, there was abundant evidence for linkage equilibrium between pairs of loci within the ST5-associated genotype cluster (Table 9). The complete allelic profiles for these clusters are shown in Appendix A.

## 4. Discussion

This study analyzed the genetic structure of geographic populations of the *C. neoformans* complex based on published multilocus sequence data. We focused on 296 STs with available geographical data. Our analyses included a robust global population of CNSC, with 4200 isolates originating across 31 countries and four continents (South and North America combined as one continent). Of the 296 sequence types with geographical data, 244 (82%) were sampled from a single continent, with 24 of these 244 STs found across multiple countries within a continent. The remaining 18% STs were distributed across multiple continents, with 28 STs (9%) found in two continents, 9 STs (3%) among three continents and 15 STs (5%) across all four continents. The broad distributions of multiple sequence types across multiple continents and countries are consistent with the recent and frequent gene flow in CNSC. Contemporary factors such as wind, animal and human migrations and other anthropogenic activities could all have facilitated the dispersals of genes and genotypes, causing wide distributions of certain genotypes [8,59,60,61].

However, our population genetic analyses revealed statistically significant differentiations among continental and national populations of CNSC. At the whole CNSC level, the observed genetic differentiations were contributed by differences in the distributions of the four molecular types and by the localized clonal expansion of specific sequence types. Indeed, evidence for the clonal expansion of specific genotypes was found for all molecular types. For example, the most abundantly collected ST in the analyzed data was ST5, represented by 1332 isolates. The second most abundant was ST93, represented by 460 isolates. Even though both were found in all four continents, ST5 and ST93 were mainly found in Asia (1211 of the 1332 isolates) and the Americas (224 of the 460 isolates), respectively. Among the serotype D (VNIV) isolates, ST160 was the most recorded sequence type, representing 78 isolates across three continents. Localized clonal expansion can significantly skew the allele and genotype frequencies and contribute to observed genetic differences among geographic populations [62]. Thus, we analyzed clone-corrected samples where only one representative from each country was included for analyses. Using the clone-corrected samples, the amount of contribution due to geographic separation to the total genetic variance reduced by 70–93% at the continental level and by 60–93% at the country level for the total CNSC, the serotype A and the serotype D samples, respectively. This result is consistent with the presence of indigenous genetic variations within most national and continental populations, likely due to historical differentiations.

Interestingly, out of the 529 isolates representing 109 STs collected within Africa, only 2 isolates belonged to serotype D (VNIV). Our meta-analysis was consistent with an earlier study that analyzed the molecular types of 505 isolates from Africa and found that none of them belonged to molecular type VNIV [61]. However, these results do not mean that VNIV is unimportant in Africa. For example, one study that analyzed 252 isolates from South Africa identified 5 cases of molecular type VNIV [36]. However, those isolates were not genotyped using MLST. At present, Africa accounts for the greatest global burden of HPC infection [63,64] and contains the most genetically diverse population of serotype A [36], including the relatively frequent distributions of both mating types. The high genetic diversity of serotype A strains in Africa has led to the “Out of Africa” hypothesis for the origin and spread of serotype A [65]. Interestingly, four STs from Africa that were originally identified as belonging to VNI were clustered to the basal clade of VNIV (Figure 1). These STs likely represent some of the ancestor genotypes of VNIV or recent hybrids between the VNI and VNIV strains.

Among the 72 serotype D (VNIV) STs, 65 (90%) were represented in Europe. In comparison, only 23% of serotype A STs were found in Europe. The results are consistent with multiple studies reporting a relatively high prevalence of serotype D within Europe [66]. The relatively broad distributions of both serotype A and serotype D strains in Europe are likely the main contributors to the frequent observations of serotype AD hybrids within Europe [66]. 

Although there is an abundant *C. neoformans* population in North America [67,68], only 12 isolates have been analyzed using the ISHAM MLST scheme. Such a lack of MLST data from North America was not due to the lack of samples for analyses. Indeed, between 1992 and 1994, the US CDC conducted a large-scale surveillance of the agents of cryptococcosis [69]. Those isolates were genotyped using random amplified polymorphic DNA and/or multilocus enzyme electrophoresis, the commonly used molecular markers at that time, revealing abundant genetic variations, including at least three independent hybridizing events between serotypes A and D [8,59,69,70]. However, those strains, as well as many strains isolated afterwards from North America, have not been genotyped using the ISHAM MLST scheme, which was published in 2009. It would be very interesting to analyze the North American population of *C. neoformans* and to compare them with those from other parts of the world. In contrast to the lack of MLST data from North America, there is a large representation of *C. neoformans* from China, making that population of CNSC one of the best for understanding fine-scale spatial and temporal structures of CNSC.

In this study, we performed a phylogenetic analysis of all STs based on their concatenated DNA sequences. Overall, the phylogenetic results were consistent with the molecular type designations for most isolates based on PCR fingerprinting, AFLP and/or PCR-RFLP of the URA5 gene fragment. However, we found that the placements of a small number of STs in the phylogenetic trees were inconsistent with their original molecular type designations (Figure 1). Such inconsistencies were found among molecular types within serotype A as well as between serotype A and serotype D. Among these inconsistently placed STs, 15 contained mixtures of alleles from serotype A and serotype D (Table 4). These STs were likely recombinants derived from the hybridization between serotypes A and D strains. After hybridization, either meiotic or mitotic recombination could have led to a loss of heterozygosity (LOH) to generate the haploid recombinant genotypes observed here. Indeed, LOH in serotype AD hybrids has been observed through both meiosis and mitosis, with environmental stress facilitating LOH [71,72]. The observed recombination is also consistent with the results from nuclear and mitochondrial genome phylogenetic comparisons, where the mitochondrial genome-based phylogeny showed several differences with that based on nuclear genome-based phylogeny within CNSC [73]. Furthermore, some of these STs had distinct alleles and showed significant divergence with the main serotype A and the main serotype D genotypes and clades, a result suggesting the existence of distinct novel lineages within CNSC (Figure 1, Appendix A). Finally, our phylogenetic analyses successfully placed 51 previously undefined STs into the phylogenetic framework and revealed their possible origins. The presence of these hybrids, as well as many intermediate STs with ambiguous molecular type assignments in the phylogeny, supports the continued use of CNSC for this group of fungal pathogens [9].

CNSC has been shown to reproduce predominantly asexually in nature, but evidence for recombination has been reported for both the serotype A and serotype D populations [28,74,75,76]. Sexual reproduction can accelerate adaptation to diverse environments and remove deleterious mutations more effectively than asexual reproduction [77]. Across all four major molecular types (VNI, VNII, VNB and VNIV) as well as two subtypes of VNB (VNBI and VNBII), we found clear evidence of recombination within each. Significantly, evidence for recombination was also found in two presumed “clonal ST clusters” within VNI. The observed results suggest that, even in geographic populations of CNSC dominated by a few STs, recombination is still possible. Such recombination could be achieved through same-sex mating or opposite-sex mating, as suggested previously when regional populations of CNSC were found to contain evidence of recombination, despite only one mating type (MATα) being found in those analyzed samples [74,75]. 

In conclusion, analyses of the published MLST data for CNSC allowed us to quantify the genetic diversity within and among geographic populations of this important human pathogenic yeast. Our analyses revealed evidence for historical geographic differentiations of CNSC, both at the whole CNSC level as well as within the populations of serotypes A and D. Not surprisingly, evidence for the clonal expansion of many STs was found at both the local population level as well as across countries and continents, suggesting the potentially important roles of recent anthropogenic activities in the dispersals of alleles and genotypes of CNSC. Importantly, we found evidence for recombination within all molecular types, including at least two presumed “clonal ST clusters” within VNI. The results indicate the diverse methods of CNSC reproduction in nature. While a large number of isolates were analyzed in this study for population genetic patterns, about 55% of the 657 STs in the ISHAM MLST database were not included for geographic structure analyses due to the lack of geographic location information associated with these 361 STs. In the future, authors should be required to submit the metadata associated with each isolate and each sequence type that they publish in their MLST study. Additional information on these STs, as well as more MLST data, especially from under-reported regions such as North America, will provide a more comprehensive understanding of the global population structure of CNSC and help develop more realistic models of global cryptococcal threat predictions and management strategies against cryptococcosis [78].

## Figures and Tables

**Figure 1 genes-13-02045-f001:**
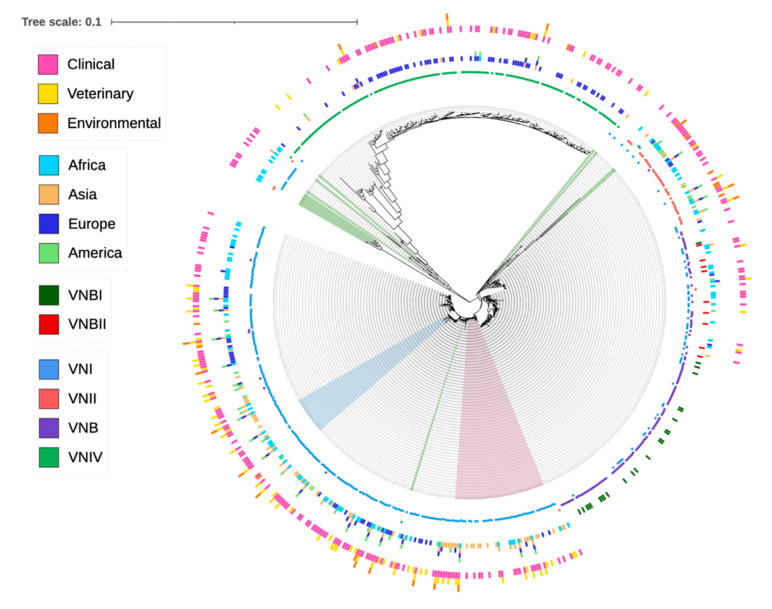
Phylogenic tree showing the relationships among 657 sequence types (ST) of the *C. neoformans* species complex. A total of 3 large circles and 10 small circles were used to display the metadata associated with each ST. The outermost large circle containing three small circles shows ecological niches, with each colored bar corresponding to one ecological niche. The middle large circle containing four small circles shows the continental distributions of each ST, with each colored bar corresponding to one continent. The next circle inside containing two colored bars corresponds to strains that were identified as VNBI (green) and VNBII (red) by authors who reported these STs. The next four small circles correspond to four original molecular types (VNI, VNII, VNB and VNIV) assigned for each ST, with each small circle corresponding to one molecular type and the presence of the ST in the molecular type indicated by a colored bar stick. Legends on the left indicate the correspondence between each color and the metadata of each ST. STs shaded in green are those showing significant deviations from their original four main molecular type assignments. STs shaded in pink and blue correspond to those closely related to ST5 and ST93, respectively, which were separately analyzed for evidence of recombination.

**Table 1 genes-13-02045-t001:** Summary of the geographic distributions of isolates of the *C. neoformans* species complex genotyped using the ISHAM multilocus sequence typing scheme.

Region	*n*	%	Region	*n*	%	Region	*n*	%	Region	*n*	%
Asia	2600	61.9%	Africa	529	12.6%	Europe	601	14.3%	South America	457	10.9%
China	1216	28.95%	South Africa	268	6.38%	France	226	5.38%	Brazil	318	7.57%
Thailand	524	12.48%	Uganda	241	5.74%	Italy	151	3.6%	Colombia	88	2.10%
India	380	9.05%	Nigeria	12	0.29%	Germany	145	3.45	Peru	51	1.21%
Vietnam	136	3.24%	Libya	3	0.07%	Turkey	41	0.98%			
Japan	119	2.83%	Tanzania	3	0.07%	Greece	24	0.57%			
Saudi Arabia	75	1.79%	Congo	1	0.02%	Spain	10	0.24%	North America	13	0.3%
South Korea	50	1.19%	DRC	1	0.02%	Cyprus	4	0.10%	USA	12	0.29%
Indonesia	44	1.05%							D. Republic	1	0.02%
Iran	30	0.71%									
Kuwait	15	0.36%									
Qatar	8	0.19%									
Malaysia	3	0.07%									

*n*: total isolates, %: percent of isolates of the total recorded in CNSC.

**Table 2 genes-13-02045-t002:** Summary distribution of 296 sequence types of the *C. neoformans* species complex among continents and ecological niches.

Distribution Patterns	Specific Continent(s)/Ecological Niche(s)	Number of Sequence Types	Number of Isolates
Geographic			
In all four continents		15	2675
In three continents only			
	Asia + Africa + Europe	1	207
	Asia + Africa + America	4	353
	Africa + America + Europe	1	4
	Asia + Europe + America	3	92
In two continents only			
	Asia + Africa	8	24
	Asia + Europe	6	197
	Asia + America	3	28
	Africa + Europe	3	23
	Africa + America	4	13
	Europe + America	4	27
In one continent only			
	Asia	50	190
	Africa	73	107
	America	19	41
	Europe	102	219
Ecological niches			
In all three niches		14	2550
In two niches only			
	Clinical + Veterinary	3	8
	Clinical + Environmental	32	1026
	Veterinary + Environmental	0	0
In one niche only			
	Clinical	184	380
	Veterinary	11	16
	Environmental	40	84

**Table 3 genes-13-02045-t003:** Allelic variation among the seven loci used for the multilocus sequence typing of the *C. neoformans* species complex. The number of overlapping alleles at each locus between the serotype A and serotype D strains was calculated based on the assigned molecular types by the MLST database.

Gene	Gene Name	Chromosome Location	Length (bp)	Total Allele Number in CNSC	Allele Number in Serotype A	Allele Number in Serotype D	Number of Overlapping Alleles between A and D
*CAP59*	Capsular-associated protein	1	560	55	46	18	9
*GPD1*	Glyceraldehyde-3-phosphate dehydrogenase	7	543–546	45	38	15	8
*IGS1*	Ribosomal RNA intergenic spacer	2	685–730	93	75	32	14
*LAC1*	Laccase	8	469–473	50	40	22	12
*PLB1*	Phospholipase	12	517–543	44	38	18	12
*SOD1*	Cu, Zn superoxide dismutase	5	526–543	68	58	19	9
*URA5*	Orotidine monophosphate pyrophosphorylase	8	636–652	57	46	24	13

**Table 4 genes-13-02045-t004:** Allelic profiles of STs with intermediate and inconsistent molecular type assignments between the original assignments in the MLST database and those based on the seven-gene phylogeny in Figure 1. Fifteen STs with mixed background colors are likely hybrids of serotypes A and D.

VNIV	ST	*CAP59*	*GPD1*	*IGS1*	*LAC1*	*PLB1*	*SOD1*	*URA5*
	254	9	1	14	6	4	1	16
	266	5	22	34	19	10	23	18
	355	22	5	1	19	3	17	20
	489	1	1	1	1	1	27	1
	521	27	13	12	6	9	8	13
	538	54	42	1	5	38	60	55
VNI								
	210	4	11	56	6	6	1	3
	224	9	11	57	6	6	33	2
	225	13	1	55	11	4	36	1
	249	34	1	55	27	4	7	1
	259	24	23	28	3	2	1	2
	263	9	11	57	6	6	27	2
	326	9	23	60	23	16	1	40
	345	5	22	34	6	10	23	3
	353	22	5	32	19	3	1	1
	354	37	5	32	19	3	1	1
	358	24	1	32	3	3	1	1
	365	22	6	32	19	3	1	1
	366	25	2	60	23	16	1	1
	651	1	1	60	3	14	1	34
VNII								
	221	8	10	58	8	2	3	11
	222	8	10	58	8	12	3	11
	363	9	22	24	19	19	1	33

The first column represents the original molecular type assignment found in the ISHAM MLST database. Alleles highlighted in blue represent those in the serotype A cluster, while those highlighted in yellow represent alleles in the serotype D cluster.

**Table 5 genes-13-02045-t005:** Analysis of molecular variance at the continental level.

	None-Clone-Corrected				Clone-Corrected			
	df	MS	Est.Var	%	df	MS	Est.Var	%
Total Population								
Among Continents	3	513.8	0.64	28% ***	3	14.5	0.1	4% ***
Within Continents	4196	1.7	1.65	72% ***	382	2.8	2.77	96% ***
Total	4199	2	2.3	100%	385	2.9	2.90	100%
Serotype A								
Among Continents	3	298.5	0.42	22% ***	3	5.2	0.04	2% ***
Within Continents	3895	1.5	1.5	78% ***	275	2.5	2.51	98% ***
Total	3898	1.7	2.93	100%	278	2.6	2.54	100%
Serotype D								
Among Continents	3	15.6	0.35	16% ***	3	3.5	0.12	4% *
Within Continents	250	1.9	1.91	84% ***	75	2.6	2.65	96% ***
Total	253	2	2.3	100%	78	2.7	2.8	100%

df: degrees of freedom; MS: mean square; Est.Var: estimated variance; %: percentage of variance; * *p* < 0.05; *** *p* < 0.001.

**Table 6 genes-13-02045-t006:** Pairwise population comparison at the continental level.

	Non-Clone-Corrected			Clone-Corrected		
	Africa	Asia	Europe	Africa	Asia	Europe
Total						
Africa						
Asia	0.089 ***			0.101 ***		
Europe	0.512 ***	0.347 ***		0.019 ***	0.050 ***	
America	0.226 ***	0.165 ***	0.254 ***	0.052 ***	0.013 ***	0.012 ***
Serotype A						
Africa						
Asia	0.088 ***			0.032 ***		
Europe	0.576 ***	0.345 ***		0.015 ***	0.019 ***	
America	0.263 ***	0.136 ***	0.15 ***	0.012 ***	0.018 ***	0.000 ***
Serotype D						
Africa						
Asia	0.217 ***			0.058 *		
Europe	0.241 ***	0.151 ***		0.011 *	0.023 *	
America	0.223 ***	0.103 ***	0.197 ***	0.223 *	0.163 *	0.066 *

* *p* <0.05; *** *p* < 0.001.

**Table 7 genes-13-02045-t007:** Analysis of molecular variance at the country level.

	None-Clone-Corrected				Clone-Corrected			
	df	MS	Est.Var	%	df	MS	Est.Var	%
Total Population								
Among Countries	23	210.8	1.3	61% ***	19	14.2	0.5	17% ***
Within Countries	34,159	0.86	0.86	39% ***	495	2.3	2.3	83% ***
Total	4182	2	2.2	100%	514	2.7	2.8	100%
Serotype A								
Among Countries	22	170.1	1.2	59% ***	17	12.9	0.52	21% ***
Within Countries	3862	0.77	0.77	41% ***	378	1.9	21.9	79% ***
Total	3884	1.8	1.9	100%	395	2.4	2.4	100%
Serotype D								
Among Countries	4	25.4	0.6	26% ***	2	3.3	0.04	1%
Within Countries	234	1.6	1.6	74% ***	68	2.6	2.6	99% ***
Total	338	2	2.1	100%	70	2.6	2.6	100%

df: degrees of freedom; MS: mean square; Est.Var: estimated variance; %: percentage of variance; *** *p* < 0.001.

**Table 8 genes-13-02045-t008:** Summary of genotypic diversity and phylogenetic incompatibility based on the sequence type and molecular type information assigned by the MLST database. ** *p* < 0.01; *** *p* < 0.001.

	Number	Phylogenetic Compatibility (% of 21 Pairs)	Index of Association
Total	657	0	0.75 ***
Serotype A	441	0	0.69 ***
VNI	296	0	0.83 ***
ST5-associated	49	23.8%	0.37 **
ST93-associated	20	76.2%	0.19
VNII	43	23.8%	0.44 ***
VNB	102 *	9.5%	0.58 ***
VNBI	93 *	23.8%	0.19 ***
VNBII	59 *	9.5%	0.99 ***
Serotype D	158	0	0.30 ***

* The number of STs analyzed here for VNB (102) was based on the molecular type assignment in the MLST database. In the MLST database, only 24 and 9 STs were assigned to VNBI and VNBII, respectively. The numbers are too small for effective analyses. Instead, in our recombination analyses for VNBI and VNBII, we extracted the numbers of STs for VNBI (93) and VNBII (59) from our phylogenetic results of the concatenated genes in Figure 1.

**Table 9 genes-13-02045-t009:** Linkage disequilibria between pairs of loci in two sequence type clusters of *C. neoformans*.

ST93 Cluster	Loci	N (# of Alleles)	*CAP59*	*GPD1*	*IGS1*	*LAC1*	*PLB1*	*SOD1*	*URA5*
	*CAP59*	(1)		<0.001	<0.001	<0.001	<0.001	<0.001	<0.001
	*GPD1*	(6)	N/A		<0.001	<0.001	<0.001	<0.001	<0.001
	*IGS1*	(4)	N/A	0.0363		<0.001	<0.001	<0.001	<0.001
	*LAC1*	(5)	N/A	0.0501	0.0336		<0.001	<0.001	<0.001
	*PLB1*	(5)	N/A	0.0973	0.2239	0.2198			<0.001
	*SOD1*	(3)	N/A	0.0347	0.0058	0.1750	0.0445		<0.001
	*URA5*	(5)	N/A	0.0942	0.2193	0.2754	0.2378	0.1907	
ST5 cluster									
	*CAP59*	(7)		0.402	0.402	0.402	0.402	0.402	0.402
	*GPD1*	(6)	0.0158		0.938	0.938	0.938	0.938	0.938
	*IGS1*	(11)	0.1506	0.0844		0.002	0.002	0.002	0.002
	*LAC1*	(7)	0.3829	0.3525	0.0265		0.013	0.013	0.013
	*PLB1*	(8)	0.2611	0.2486	0.0770	0.5576		0.768	0.768
	*SOD1*	(7)	0.0650	0.0705	0.0173	0.1190	0.1314		0.024
	*URA5*	(8)	0.2110	0.0329	0.2518	0.2640	0.1588	0.1501	

*n* = number of allele types recorded for each gene for the two analyzed samples. Numbers in the bottom left part of the table indicate the linkage disequilibrium D values between pairs of loci. Numbers in the top right part of the table indicate corresponding *p* values. *p* < 0.05 indicates a rejection of the null hypothesis that the two loci are in linkage equilibrium.

## Data Availability

All data analyzed in this study are cited and are summarized in the manuscript.

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
