# Peer review of "Analyses of the Global Multilocus Genotypes of the Human Pathogenic Yeast Cryptococcus neoformans Species Complex"

_genes, 2022, doi:10.3390/genes13112045_

Round 1

Reviewer 1 Report

The research is a good statistically generated study that analyzes published MLST data. The lack of geographic information for many analyzes significantly limited the study. In addition, the fact that the data for North America is less than for other continents makes it difficult to interpret.

Suggestions;

line 367, please check the reference citation [eg. 58] ??

VNB consists of two lineages: VNBI and VNBII. Why were these lineages excluded?

Author Response

Dear Reviewer,

Thanks for your time and expertise in helping us improve our study. Following your suggestions, we have revised the manuscript accordingly. The changes within the manuscript are highlighted in yellow. Please see our specific responses below for each of your comments.

Comment #1: The research is a good statistically generated study that analyzes published MLST data. The lack of geographic information for many analyzes significantly limited the study. In addition, the fact that the data for North America is less than for other continents makes it difficult to interpret.

Response: Yes, we agree with your assessment and have further highlighted the need for submitting isolate metadata (lines 482-485) and for obtaining additional MLST data, especially from under-reported areas such as North America (lines 418-429; 482-485).

Suggestions;

Comment #2: line 367, please check the reference citation [eg. 58] ??

Response: Done. The sentence has been changed slightly and the reference list has been updated (lines 376-378).

Comment #3: VNB consists of two lineages: VNBI and VNBII. Why were these lineages excluded?

Response: Thanks for bringing this up. In the original version, we didn't separate the two subtypes of VNB in our analyses because the MLST database contained VNBI and VNBII classification for only 24 and 9 STs respectively. Both were too small for population genetic analyses. In the revised version, we have now further analyzed the data. Specifically, first, in Figure 1, the VNBI and VNBII designations for the 24 and 9 STs are now added. Second, based on the concatenated gene tree, we were able to identify a total of 93 VNBI STs and 59 VNBII STs in the database and these STs were then used for recombination analyses (highlighted in Table 8). Interestingly, evidence for recombination was found in both VNBI and VNBII. The descriptions of the new results are highlighted in yellow in section 3.5. The information was also added to the abstract.

Please see the attached pdf for the specific changes (highlighted in yellow) in the revised version of the manuscript.

Yours Sincerely,

JP

Reviewer 2 Report

Using the global data available from LST typing of Cryptococcus, the authors aim to estimate the overall genetic variation and determine if recombination plays a role in the diversity observed within molecular types of cryptococcus isolates. 

The only minor comment I have for the manuscript is:

1. Consider moving Table 2 to supplementary material, and introduce a more reduced Table within the main document which summarizes the distribution of the 296 MLST with known geographic information

Author Response

Dear Reviewer,

Thank you for your time and expertise in helping us improve the manuscript. We have now made the suggested changes to Table 2. The detailed geographic and ecological distributions of each of the analyzed sequence types are included in Supplementary Tables S1 and S2.

Please see attached the modified version that included responses to both your and the other reviewer's suggestions. The changes are highlighted in the pdf.

Sincerely Yours,

JP
